# A 10-Minute “Mix and Read” Antibody Assay for SARS-CoV-2

**DOI:** 10.3390/v13020143

**Published:** 2021-01-20

**Authors:** Juuso Rusanen, Lauri Kareinen, Lev Levanov, Sointu Mero, Sari H. Pakkanen, Anu Kantele, Fatima Amanat, Florian Krammer, Klaus Hedman, Olli Vapalahti, Jussi Hepojoki

**Affiliations:** 1Department of Virology, Medicum, Faculty of Medicine, University of Helsinki, 00014 Helsinki, Finland; lauri.kareinen@helsinki.fi (L.K.); lev.levanov@helsinki.fi (L.L.); klaus.hedman@helsinki.fi (K.H.); olli.vapalahti@helsinki.fi (O.V.); 2Department of Veterinary Biosciences, Faculty of Veterinary Medicine, University of Helsinki, 00014 Helsinki, Finland; 3Human Microbiome Research Program, Faculty of Medicine, University of Helsinki, 00014 Helsinki, Finland; sointu.mero@helsinki.fi (S.M.); sari.pakkanen@hus.fi (S.H.P.); anu.kantele@helsinki.fi (A.K.); 4Meilahti Vaccination Research Center (MeVac), Inflammation Center, Helsinki University Hospital and University of Helsinki, 00029 Helsinki, Finland; 5Department of Microbiology, Icahn School of Medicine at Mount Sinai, New York, NY 10029, USA; fatima.amanat@icahn.mssm.edu (F.A.); florian.krammer@mssm.edu (F.K.); 6Graduate School of Biomedical Sciences, Icahn School of Medicine at Mount Sinai, New York, NY 10029, USA; 7HUS Diagnostic Center, HUSLAB, Helsinki University Hospital and University of Helsinki, 00029 Helsinki, Finland; 8Institute of Veterinary Pathology, Vetsuisse Faculty, University of Zürich, CH-8057 Zürich, Switzerland

**Keywords:** SARS-CoV-2, serology, serodiagnosis, TR-FRET, immunoassay

## Abstract

Accurate and rapid diagnostic tools are needed for management of the ongoing coronavirus disease 2019 (COVID-19) pandemic. Antibody tests enable detection of individuals past the initial phase of infection and help examine vaccine responses. The major targets of human antibody response in severe acute respiratory syndrome coronavirus 2 (SARS-CoV-2) are the spike glycoprotein (SP) and nucleocapsid protein (NP). We have developed a rapid homogenous approach for antibody detection termed LFRET (protein L-based time-resolved Förster resonance energy transfer immunoassay). In LFRET, fluorophore-labeled protein L and antigen are brought to close proximity by antigen-specific patient immunoglobulins of any isotype, resulting in TR-FRET signal. We set up LFRET assays for antibodies against SP and NP and evaluated their diagnostic performance using a panel of 77 serum/plasma samples from 44 individuals with COVID-19 and 52 negative controls. Moreover, using a previously described SP and a novel NP construct, we set up enzyme linked immunosorbent assays (ELISAs) for antibodies against SARS-CoV-2 SP and NP. We then compared the LFRET assays with these ELISAs and with a SARS-CoV-2 microneutralization test (MNT). We found the LFRET assays to parallel ELISAs in sensitivity (90–95% vs. 90–100%) and specificity (100% vs. 94–100%). In identifying individuals with or without a detectable neutralizing antibody response, LFRET outperformed ELISA in specificity (91–96% vs. 82–87%), while demonstrating an equal sensitivity (98%). In conclusion, this study demonstrates the applicability of LFRET, a 10-min “mix and read” assay, to detection of SARS-CoV-2 antibodies.

## 1. Introduction

In December 2020, the number of confirmed cases in the ongoing coronavirus disease 2019 (COVID-19) pandemic, caused by the severe acute respiratory syndrome coronavirus 2 (SARS-CoV-2), exceeded 70 million, with over 1.5 million deaths [1]. Reliable diagnostic assays are needed for management of COVID-19 patients and epidemic containment (“test, trace and isolate”). Nucleic acid tests (NAT) or antigen tests serve to detect acute SARS-CoV-2 infection, whereas antibody testing tells the past-infection and/or immunity status. Hence, antibody tests can be used for determining seroprevalences, examining vaccine responses in study settings, or finding out whether an individual needs a booster as in the case of e.g., hepatitis B vaccine. With COVID-19, antibody testing may be the key in reaching the diagnosis for a patient presenting when the viral RNA has already waned, e.g., with late thromboembolic complications or prolonged symptoms [2]. The most widespread methods in antibody detection are enzyme immunoassays (EIAs) and lateral flow assays (LFAs); the former tend to be highly specific and sensitive yet require dedicated infrastructure and labor, and deliver the results at best within hours, whereas LFAs are simple and rapid but may be of substandard diagnostic performance.

We have previously set up rapid homogeneous (wash-free) immunoassays utilizing time-resolved Förster resonance energy transfer (TR-FRET) [3,4,5,6,7,8]. For FRET to occur, a donor and acceptor fluorophore are brought to close proximity (<100 Å), allowing excitation of the donor to result in energy transfer to the acceptor, which then emits at a distinct wavelength. To reduce autofluorescent background, a chelated lanthanide donor exhibiting long-lived fluorescence is employed, allowing for time-resolved measurement (TR-FRET). We have developed a TR-FRET -based immunoassay concept termed LFRET (protein L-based time-resolved Förster resonance energy transfer immunoassay) and demonstrated its excellent diagnostic performance in detection of antibodies against Puumala orthohantavirus nucleocapsid protein, Zika virus NS1 and the autoantigen tissue transglutaminase [5,7,8]. LFRET relies on simultaneous binding to the antibody of interest of its donor-labeled antigen and of an acceptor-labeled protein L. If the patient’s serum contains antibodies against the antigen, they bring the two fluorophores to close proximity, generating a TR-FRET signal. Interestingly, a recent report describes a TR-FRET based 1-h assay for separate detection of anti-SARS-CoV-2 antibodies of different immunoglobulin isotypes [9].

SARS-CoV-2 is an enveloped (+)ssRNA virus with a non-segmented 30 kb genome and four structural proteins: spike (SP), envelope (E), membrane (M), and nucleoprotein (NP). Protruding from the viral surface are transmembrane homotrimers of SP, essential for host cell entry. The S glycoprotein is proteolytically cleaved into subunits S1 and S2, of which S1 contains the host cell receptor-binding domain (RBD), while S2 mediates fusion with the host cell membranes [10]. Like SP, the E and M proteins are located on the viral envelope, whereas NP binds the viral RNA to form a ribonucleoprotein complex that is encapsulated within the viral membrane.

Antibody responses to SARS-CoV-2 predominantly target the NP and SP. In hospitalized patients, the median time from onset of symptoms to IgA, IgM and IgG seroconversion has been observed to be 11-14 days, with almost all seroconverting by day 21 [11,12,13]. The antibody levels correlate with the severity of disease, with few patients apparently not seroconverting [12]. Moreover, a fraction of the seroconverters do not seem to generate detectable neutralizing antibodies (NAbs) [14]. The NAb response correlates with the presence of anti-SP antibodies [15,16], with most but not all NAbs targeting the RBD [17]. IgG levels to other human betacoronaviruses peak within months of infection and wane within years thereafter [18,19], and reinfections with seasonal human coronaviruses occur as early as 12 months from the previous infection [20]. As for SARS-CoV-2, the persistence of antibodies and the extent to which they provide protective immunity remains as of yet uncertain.

In this study we introduce rapid wash-free LFRET assays for detection of antibodies against SARS-CoV-2 NP and SP antigens and compare them with ELISAs and microneutralization.

## 2. Materials and Methods

### 2.1. Samples

This study included 77 serum/plasma samples from 40 individuals tested positive and four samples from four individuals tested negative for SARS-CoV-2 by RT-PCR from nasopharyngeal swab samples. The positive samples were taken at 8 to 81 days after onset of symptoms. Additionally, 48 serum samples from asymptomatic individuals with a comprehensively negative SARS-CoV-2 serology (Euroimmun IgG (EUROIMMUN Ag, Lübeck, Germany), IFA IgG virus, IFA IgG spike, microneutralization negative) and 27 seropositive samples (Euroimmun SARS-CoV-2 ELISA (IgG), Abbott Architect SARS-CoV-2 IgG (Abbott, Abbott Park, IL, USA) and microneutralization positive) were included. The data and samples were collected under research permit HUS/211/2020 and ethics committee approval HUS/853/2020 (Helsinki University Hospital, Finland).

### 2.2. Nucleic Acid Testing

Nucleic acid testing for SARS-CoV-2 was done from nasopharyngeal swab samples with either the Cobas^®^ SARS-CoV-2 test using the Cobas^®^ 6800 system (Roche Diagnostics, Basel, Switzerland), a protocol based on one previously described by Corman et al. [21], or the Amplidiag^®^ COVID-19 test (Mobidiag, Espoo, Finland).

### 2.3. Molecular Cloning

For protein expression, we acquired the ZeoCassette Vector (pCMV/Zeo) from ThermoFisher Scientific (Walham, MA, USA), and excised the Zeocin resistance gene from the vector using FastDigest EcoRI and XhoI (ThermoFisher Scientific) according to the manufacturer’s protocol. The excised gene was agarose gel purified, blunted using T4 DNA polymerase (ThermoFisher Scientific), and purified using Ampure XP beads (Beckman Coulter, Brea, CA, USA) both following the manufacturer’s protocol. The selection gene was inserted into pCAGGS/MCS and to the pCAGGS vector bearing SARS-CoV-2 S protein (described in [22,23]) gene by treating the plasmids with FastDigest SapI/LguI (ThermoFisher Scientific) according to the manufacturer’s protocol, followed by blunting and purifications as above. The insert was ligated to the plasmids using T4 DNA ligase (ThermoFisher Scientific) according to the manufacturer’s protocol, and the products were transformed into Escherichia coli (DH5a strain), followed by plating of the bacteria onto LB plates with 100 µg/mL of ampicillin and 50 µg/mL Zeocin (ThermoFisher Scientific). After overnight incubation at 37 °C, single colonies were picked and grown in 5 mL of 2 × YT medium supplemented with 100 µg/mL of ampicillin and 50 µg/mL Zeocin overnight at 37 °C. The plasmids were purified using GeneJET Plasmid Miniprep Kit (ThermoFisher Scientific), and those bearing the insert were identified by restriction digestion (FastDigest EcoRI, ThermoFisher Scientific) and agarose gel electrophoresis. For both constructs, clones with the insert in reverse and forward direction were selected for ZymoPURE II Plasmid Maxiprep Kit (ZymoResearch, Irvine, CA, USA) preparations done following the manufacturer’s guidelines. A synthetic SARS-CoV-2 NP gene under Kozak sequence and a signal sequence MMRPIVLVLLFATSALA flanked by KpnI and SgsI restriction sites was obtained from ThermoFisher Scientific. The SARS-CoV-2 NP cassette was subcloned into pCAGGS/MCS-Zeo-fwd vector and plasmid maxipreps prepared as described above.

### 2.4. Protein Expression and Purification

We initially attempted producing SARS-CoV-2 SP in Expi293F cells utilizing the Expi293 Expression System (ThermoFisher Scientific). Briefly, the pCAGGS plasmid bearing codon-optimized SARS-CoV-2 spike1 was transiently transfected into Expi293F cells as advised [22,23], except that we used spinner flasks (disposable 125 mL spinner flask, Corning, Corning, NY, USA) for the culture. Protein purification from supernatants collected at five days post transfection followed the protocol described and yielded 0.2–0.3 mg per 100 mL culture, in line with earlier reports [22,23].

Next, we transfected adherent HEK293T cells with pCAGGS-SARS-CoV-S-Zeo plasmids using Fugene HD at 3.5:1 ratio, in suspension as described [24]. The transfected cells were plated onto six-well plates, and at 48 h post transfection subjected to Zeocin selection, 150 µg/mL Zeocin in high glucose DMEM (Dulbecco’s Modified Eagle’s Medium, MilliporeSigma, St. Louis, MO, USA) supplemented with 5% fetal bovine serum (Gibco, ThermoFisher Scientific) and 4 mM L-glutamine. Two days after initiating the selection, the cells were trypsinized and transferred into fresh wells, with fresh media and antibiotics provided at 2 to 3 day intervals. Once confluent, the cells were trypsinized, counted (TC20 cell counter, Bio-Rad Laboratories, Hercules, CA, US), diluted to ~30 cells/mL, and dispensed onto 96-well plates, 100 µL per well. Once confluent, the cells were switched to serum-free FreeStyle 293 Expression Medium (ThermoFisher Scientific) with 100 µg/mL Zeocin, and incubated at 37 °C 5% CO2. At 48 h, the medium was analyzed for the presence of SARS-CoV-2 SP by dot blotting; briefly via drying 2.5 µL of the supernatant onto a nitrocellulose membrane, which then was blocked (3% skim milk in Tris-buffered saline with 0.05% Tween-20), washed, probed with rabbit anti-RBD (40592-T62, Sino Biological, Beijing, China), washed, probed with anti-rabbit IRDye800 (LI-COR Biosciences, Lincoln, NE, USA), washed, and read using Odyssey Infrared Imaging System (LI-COR Biosciences). The clone with most SARS-CoV-2 SP in supernatant, HEK293T-spike-D5, was then expanded in DMEM with 5% FBS, 4 mM L-glutamine and 100 µg/mL Zeocin, and ampouled for storage in liquid nitrogen. The HEK293T-spike-D5 cells were suspension cultured in spinner flasks in Expi293 Expression Medium (ThermoFisher Scientific) with 100 µg/mL of Zeocin, and stored in liquid nitrogen and tested for the ability to produce SARS-CoV-2 S protein in both Expi293 and FreeStyle 293 Expression Medium (both ThermoFisher Scientific). After culturing in the spinner flask for 5 to 8 days, a density of >3 × 10^6^ cells/mL was reached. The protein from the supernatant was purified as described2, with yields of 0.8–1.2 mg per 100 mL.

For production of SARS-CoV-2 NP, Expi293F cells were transfected with pCAGGS-SARS-CoV-2-NP-Zeo using Fugene HD; briefly, with 100 µg of the plasmid diluted into 10 mL of OptiMEM (MilliporeSigma), and 350 µL of Fugene HD added, followed by mixing and 15 min incubation at room temperature, after which the plasmid mix was added onto Expi293F cells in Expi293 Expression Medium at 2.5 × 10^6^ cells/mL. After four days the supernatant was collected, and the protein purified (yield ~1 mg per 100 mL culture) as described for S protein2. The remaining cells were treated with 0.25% Trypsin-EDTA (MilliporeSigma) to remove dead cells, and after two washes were put back into the spinner flask with fresh Expi293 Expression Medium supplemented with 100 µg/mL of Zeocin. Eventually a population of cells started to proliferate (Expi-NP-zeo cells), and were aliquoted in liquid nitrogen. The protein production and purification occurred as described above with yields of ~1 mg per 100 mL culture.

### 2.5. Protein Labeling

We labeled the SARS-CoV-2 SP and NP with the donor fluorophore europium (Eu) using QuickAllAssay Eu-chelated protein labeling kit (BN Products and Services Oy, Turku, Finland) according to the manufacturer’s instructions to generate Eu-labeled SP (Eu-SP) and NP (Eu-NP). We also labeled recombinant protein L (Thermo Scientific) with the acceptor fluorophore Alexa Fluor 647 to generate AF647-labeled protein L (AF-L), as reported [5]. IgG-free bovine serum albumin (BSA) was from Jackson ImmunoResearch Inc. (West Grove, PA, USA).

### 2.6. TR-FRET Assays

The LFRET assay was done as described [5] and as depicted in the flowchart in Figure 1. For calculating the relative TR-FRET signal increase, we here replace the pool of negative sera with TBS-BSA (50mM Tris-HCl, 150mM NaCl, pH 7.4, 0.2% BSA). The relative signal increase was converted into arbitrary counts by multiplying with 294.6111 (SP) or 380.0231 (NP). To establish LFRET assays for SP and NP, we optimized the component concentrations by cross-titration using three positive and three negative sera. For detection of anti-NP antibodies, we found the optimal on-plate conditions to be serum 1/25, AF-L 500 nM and Eu-NP 5 nM. Likewise, for detection of anti-SP antibodies, we found on-plate conditions of serum 1/100, AF-L 250 nM and Eu-SP 5 nM optimal. After combining the reagents, TR-FRET counts were measured at 0, 7, 15, 22, 30, 45 and 60 min with Wallac Victor2 fluorometer (PerkinElmer, Waltham, MA, USA) and normalized as described [3].

### 2.7. Enzyme Linked Immunosorbent Assays (ELISAs)

We set up the SARS-CoV-2 SP ELISA as described [23] with the following amendments. We coated the plates (ThermoFisher Scientific NUNC-immuno 446,442 polysorp lockwell C8) with 50 µL/well of antigens diluted 1 µg/mL in 50 mM carbonate-bicarbonate buffer pH 9.6 (Medicago AB, Uppsala, Sweden). As secondary antibodies we made use of polyclonal rabbit anti-human IgA-horeseradish peroxidase (HRP), -IgM-HRP, and -IgG-HRP (all from Dako, Jena, Germany) at respective dilutions of 1:5000, 1:1500, and 1:6000. 1-Step Ultra TMB-ELISA Substrate Solution (ThermoFisher Scientific) served for the colorimetric reaction that was terminated by 0.5 M sulphuric acid (Fluka, Buchs, Switzerland), and the absorbances recorded (HIDEX Sense, Hidex Oy, Turku, Finland) at 450 nm. The N protein ELISA followed the same protocol.

### 2.8. Microneutralization

For the SARS-CoV-2 microneutralization assay we first cultured Vero E6 cells on 96-well plates (ThermoFisher Scientific) overnight at +37°C in 2% MEM (Eagle Minimum Essential Media (MilliporeSigma) supplemented with 2% inactivated fetal bovine serum (ThermoFisher Scientific), 2 mM L-glutamine (ThermoFisher Scientific), 100 units penicillin, and 100 µg/mL streptomycin (MilliporeSigma)). The following day we made a two-fold dilution series (1:20 to 1:1280) of the serum samples in 2% MEM and combined 50 µL of each dilution with 50 µL of virus (1000 plaque forming units (pfu)/mL in 2% MEM). The serum-virus mixes were kept for 1h at +37 °C. The cells were inoculated with the serum-virus mixes and grown at +37 °C. After 4 days, the cultures were formalin-fixed, stained with crystal violet and the neutralization titers recorded. 

## 3. Results

### 3.1. LFRET Incubation Time, Cutoff Values and Performance

LFRET assays for SARS-CoV-2 SP and NP were set up using Eu-labeled in-house antigens and AF-labeled protein L. First, the assay conditions were optimized separately for SP and NP using three known anti-SP/-NP ELISA-positive and three known anti-SP/-NP ELISA-negative samples (included in the full 129-sample panel). The remaining 123 samples were tested in the optimized conditions. For detection of both anti-SP and -NP antibodies, measurement at 7 min was found optimal.

Cutoffs for both anti-SP and -NP LFRET were set by measuring LFRET signals relative to buffer in 48 samples tested negative by anti-SP and -NP ELISA. The average plus four standard deviations was set as cutoff: 228.37 + 4 × 27.59 = 338.76 counts for anti-SP and 220.94 + 4 × 27.73 = 331.86 counts for anti-NP LFRET.

Performances of the anti-SP and anti-NP LFRET assays were then determined with the 129 samples including 77 sera or heparin/EDTA plasmas from 44 individuals with a previous RT-PCR-confirmed SARS-CoV-2 infection, four samples from four individuals negative for SARS-CoV-2 by both RT-PCR and serology, and 48 samples from individuals with a comprehensively negative SARS-CoV-2 serology. The sensitivities and specificities of SARS-CoV-2 anti-SP and anti-NP LFRET in detection of PCR-positive individuals were 86% and 100%, and 76% and 100%, respectively. The combined anti-SP/-NP LFRET sensitivity and specificity were 90% and 100%; if either anti-SP or -NP LFRET was positive, the composite result was considered positive (Table 1). The development of LFRET signals over time among patients with follow-up samples available is shown in Appendix A.

### 3.2. ELISAs and Microneutralization

In order to compare the performance of LFRET with classical serology, we tested the set of samples described above with SARS-CoV-2 anti-SP and anti-NP IgA, IgM and IgG ELISAs as well as with SARS-CoV-2 microneutralization. The panel of seronegatives was excluded from anti-NP IgA and IgM, as well as from anti-SP IgM ELISAs. Altogether 107 samples underwent microneutralization, including 64 samples from RT-PCR positive patients and 43 seronegative samples. Microneutralization titers of ≥20 were considered positive.

The ELISA cutoffs were set at average plus four standard deviations of absorbances measured from 14 serum samples from SARS-CoV-2 seronegative Department staff members.

The sensitivities and specificities of ELISAs for anti-SP IgA, IgG and IgM in samples from SARS-CoV-2 RT-PCR-positive individuals were 91% and 98%, 90% and 100%, and 66% and 100%, respectively. The corresponding sensitivities and specificities of anti-NP ELISAs were 75% and 100% (IgA), 92% and 94% (IgG), and 16% and 100% (IgM), respectively (Appendix A). Pearson correlation between anti-SP and anti-NP IgG ELISAs was 0.90, 0.79 between IgM ELISAs and 0.31 between IgA ELISAs.

### 3.3. Comparison of LFRET, ELISA and Microneutralization

Comparison between the LFRET signals and ELISA absorbances is presented in Figure 2. For anti-NP antibodies, the correlation between LFRET and IgA or IgM ELISA results was low (R = 0.25 for IgA and R = 0.13 for IgM). With IgG ELISA a stronger correlation of R = 0.62 was seen, apparently hampered by saturation of the ELISA signal. For anti-SP-antibodies, correlations between IgA, IgG and IgM ELISAs were R = 0.52, R = 0.62 and R = 0.56, respectively. Higher LFRET signals were seen in samples from patients with severe disease, especially in anti-SP LFRET and when samples taken less than two weeks from onset were excluded (Appendix A). The agreement between anti-NP ELISA and LFRET was 88–89%, and that between anti-SP ELISA and LFRET 96–98% (Appendix A). The samples representing discordance between PCR, LFRET and/or ELISA are detailed in Appendix A.

The LFRET and ELISA results are compared with microneutralization titers in Figure 3. Higher neutralization titers were observed in samples from hospitalized individuals. Sensitivities and specificities of LFRET and ELISA in samples identified as microneutralization-positive or -negative are shown in Appendix A.

We also assessed the two LFRET assays using receiver operating characteristic (ROC) curves. With the assumption that a positive result in either IgM, IgG or IgA ELISA for a given sample signifies “true” positivity, the respective areas under the curve (AUCs) for both the anti-NP and the anti-SP LFRET assays were very high, 0.94 and 0.97 (Appendix A).

### 3.4. Comparison of LFRET with Commercial Assays

To assess agreement between LFRET and commercial antibody assays, we tested in anti-NP- and anti-SP-LFRET 27 samples previously determined positive by both Euroimmun and Abbott Architect SARS-CoV-2 IgG assay using manufacturer-defined cutoffs. Moreover, the 48 negative samples were tested with the Euroimmun assay. Agreement between anti-SP-LFRET and the commercial assays was 100%, whereas with anti-NP-LFRET it was 93% (Appendix A).

## 4. Discussion

We set up rapid LFRET immunoassays for detection of anti-SARS-CoV-2 SP and NP antibodies to identify individuals exhibiting an immune response against SARS-CoV-2. Management of both COVID-19 patients and the ongoing pandemic at the population level calls for accurate diagnostic tools applicable in various settings, including resource poor areas without central laboratory facilities. Antibody assays allow for detection of individuals past the initial infection phase as well as for assessment of a possible vaccine response.

We have previously employed LFRET in diagnosis of viral and autoimmune diseases. Here, we reduced the incubation time from 20 to 7 min. The simple and rapid “mix and read” workflow of the assay could allow faster turnaround time from sample arrival to results as well as higher throughput compared to the currently popular ELISAs. Moreover, the ease and speed of performing LFRET makes it feasible for use in diverse environments, including point-of-care and limited-resource settings.

Combined anti-SP/-NP LFRET (i.e., if either assay is positive, the composite result is positive) was equal to anti-SP IgG ELISA in terms of sensitivity (90%) and specificity (100%) in identification of RT-PCR -positive individuals (Table 1). In hospitalized patients, the sensitivities of both anti-SP and anti-NP LFRET reached 100% by two weeks from symptom onset, importantly for clinical use. The LFRET signals of hospitalized COVID-19 patients exceeded those of non-hospitalized (Figure 2 and Appendix A), in line with previous studies showing higher antibody levels in patients with severe clinical presentation [12,15]. In follow-up, the LFRET signals first showed a rapid rise within three weeks from onset and thereafter plateaued or slowly declined (Appendix A).

The agreement between ELISA and LFRET was ~90% for anti-NP -antibodies and >95% for anti-SP -antibodies (Appendix A), and respective AUCs were 0.94 and 0.97 (Appendix A). With the anti-NP and anti-SP LFRET vs. IgG ELISA results combined, the overall agreement between the methods was 96% (124/129 samples). A closer look at the discordance (Appendix A) shows three samples (65, 72 and 86) from PCR-positive individuals who remained seronegative in both LFRET and ELISA, likely due to early sampling (Appendix A). Four samples (71, 70, 7 and 24) were negative in LFRET but positive in PCR and ELISA: The first two were taken 8 and 13 days post onset of symptoms and positive in anti-NP IgA ELISA, suggestive of early IgA seroconversion, a phenomenon observed previously [25]. The other two were taken 4 and 9 weeks after onset from non-hospitalized patients positive in anti-SP and -NP IgG and/or IgA ELISAs. These ELISA reactivities were weak (Figure 2e), suggesting that the negativity in LFRET might reflect lower analytical sensitivity. Two samples (82, 92) were negative in LFRET and RT-PCR but positive in anti-NP IgG ELISA. Additionally, two samples (103, 121) were negative in LFRET and MNT, but positive in either anti-NP IgG ELISA or anti-SP IgA ELISA. No false positives were observed in LFRET. Agreement rates between commercial assays and anti-SP-/anti-NP-LFRET were 100% and 93%, respectively (Appendix A).

In microneutralization, all but one of the reactive samples were also positive in anti-SP LFRET, IgG and IgA ELISAs and anti-NP IgG ELISA (Figure 3), the exception being an ICU patient sampled 13 days after onset with an MNT titer of 20 and positive anti-NP IgA ELISA (Appendix A, sample 70). Some seroconverters did not exhibit a detectable NAb response, as observed previously [14]. Interestingly, the specificities of the LFRET assays in identification of the non-neutralizing individuals as negatives (91% for SP and 96% for NP) were higher than those of IgG ELISAs (87% for SP and 82% for NP) (Appendix A). This may be due to lower analytical sensitivity of LFRET, as the undetectable neutralization could result from lower overall levels of anti-SARS-CoV-2 antibodies in the LFRET-negative but ELISA-positive samples. Nevertheless, among the assays evaluated, anti-SP LFRET demonstrated the best overall performance in identification of samples containing NAbs, with a sensitivity of 98% and a specificity of 91%.

Our study has some limitations. Our SARS-CoV-2 -positive samples originated from symptomatic patients. Individuals with asymptomatic infection may mount a lower antibody response [26], whereby the sensitivity of LFRET among such individuals might be lower. Moreover, we did not examine antibodies against the widely circulating coronaviruses OC43, HKU1, NL63 and 229E, which could cross-react in the SARS-CoV-2 LFRET and reduce its specificity. However, the RT-PCR and neutralization results strongly indicated that the observed antibody responses were SARS-CoV-2 -specific.

In conclusion, this study demonstrates the applicability of the LFRET approach to detection of SARS-CoV-2 antibodies. While the new approach in sensitivity and specificity appears to parallel ELISA, it is as rapid and easy to perform as LFA, requiring only combination of the diluted sample with a reagent mix and reading the result after 7 min. In prediction of neutralization capacity, anti-SP LFRET outperformed ELISA in specificity, at equal sensitivity.

## Figures and Tables

**Figure 1 viruses-13-00143-f001:**
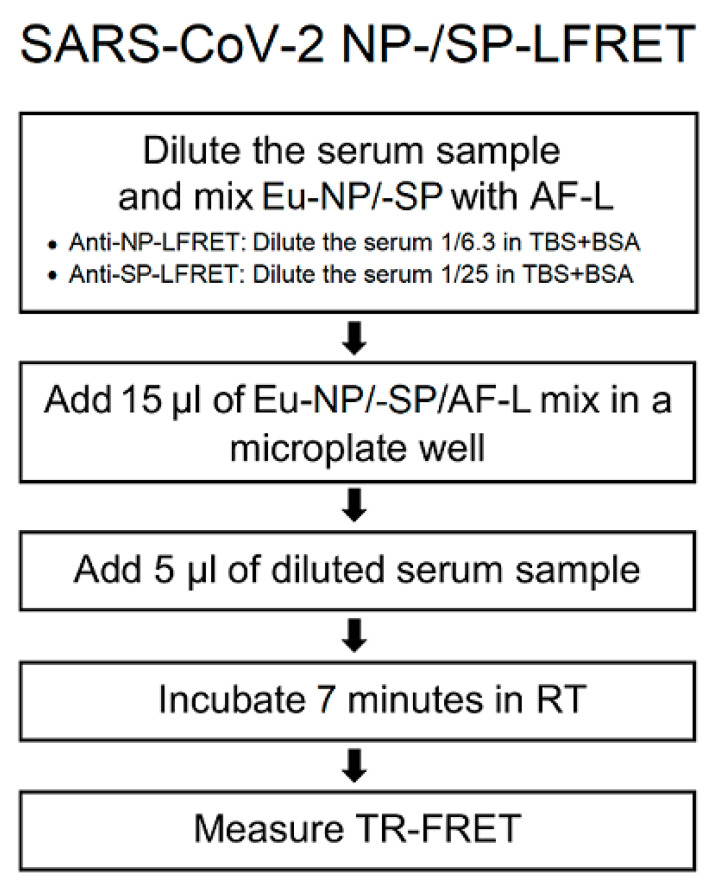
Simplified protocol for SARS-CoV-2 NP and SP LFRET assay. Eu-NP/-SP = Europium-labeled nucleoprotein/spike glycoprotein. AF-L = Alexa Fluor™ 647 -labeled protein L. TR-FRET = time-resolved Förster resonance energy transfer. RT = room temperature. TBS+BSA (50 mM Tris-HCl, 150 mM NaCl, pH 7.4, 0.2% BSA) was used for all dilutions. On-plate dilutions were 5 nM Eu-NP/500 nM AF-L/serum 1/25 for anti-NP and 5 nM Eu-SP/250 nM AF-L/serum 1/100 for anti-SP LFRET. For further details see the prior publication [5].

**Figure 2 viruses-13-00143-f002:**
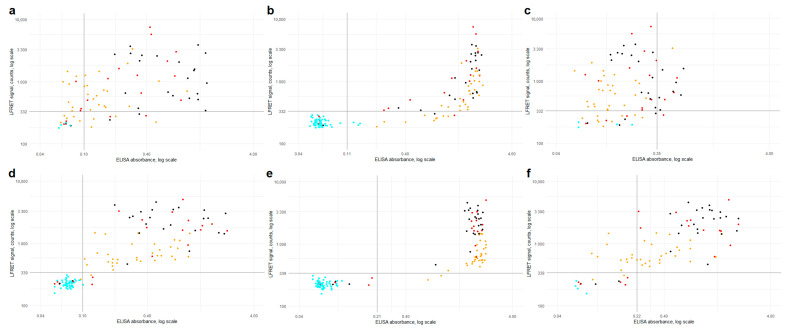
ELISA (*x*-axis) vs. LFRET (*y*-axis) results by disease severity. (**a**) Anti-NP IgA ELISA vs. anti-NP LFRET (N = 81, R = 0.25). (**b**) anti-NP IgG ELISA vs. anti-NP LFRET (N = 129, R = 0.62). (**c**) anti-NP IgM ELISA vs. anti-NP LFRET (N = 81, R = 0.13). (**d**) anti-SP IgA ELISA vs. anti-SP LFRET (N = 129, R = 0.53). (**e**) anti-SP IgG ELISA vs. anti-SP LFRET (N = 129, R = 0.62). (**f**) anti-SP IgM ELISA vs. anti-SP LFRET (N = 81, R = 0.56). Color of the dot indicates SARS-CoV-2 PCR result and disease severity: cyan = PCR negative; yellow = non-hospitalized, PCR-positive; red = non-ICU hospitalized, PCR positive; black = hospitalized in ICU, PCR positive. Horizontal and vertical black lines indicate LFRET and ELISA cutoffs. On the *x*-axis, ELISA absorbance on a logarithmic scale and on the *y*-axis, LFRET signal on a logarithmic scale. SP = spike glycoprotein. NP = nucleoprotein. LFRET = protein L–based time-resolved Förster resonance energy transfer immunoassay. ELISA = enzyme immunoassay. R = Pearson’s correlation coefficient.

**Figure 3 viruses-13-00143-f003:**
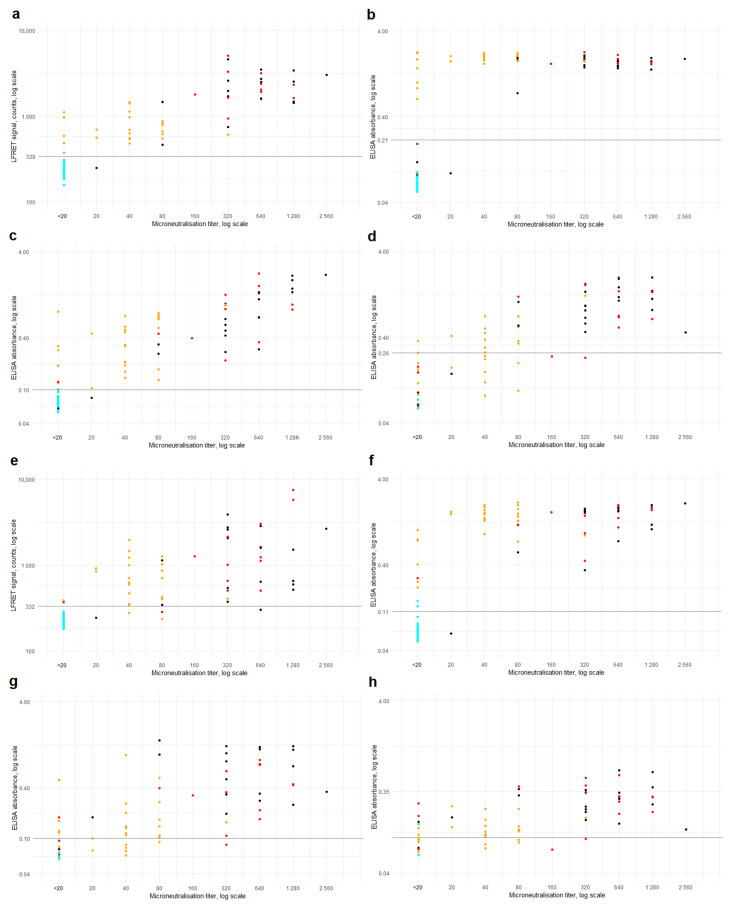
Microneutralization vs. LFRET and ELISA. Microneutralization titers are on the *x*-axis and LFRET signal or ELISA absorbance on the *y*-axis. Logarithmic scale is used on both axes. (**a**) Microneutralization titer vs. anti-SP LFRET signal (N = 107, *ρ* = 0.87). (**b**–**d**) Microneutralization titer vs. anti-SP IgG, IgA and IgM ELISA (N = 107, 107 and 67, *ρ* = 0.68, 0.86 and 0.81). (**e**) Microneutralization titer vs. anti-NP LFRET signal (N = 107, *ρ* = 0.83). (**f**–**h**) Microneutralization titer vs. anti-NP IgG, IgA and IgM ELISA (N = 107, 67 and 67, *ρ* = 0.81, 0.69 and 0.61). Color of the dots indicate SARS-CoV-2 PCR result and disease severity: cyan = PCR negative; yellow = non-hospitalized, PCR-positive; red = non-ICU hospitalized, PCR positive; black = hospitalized in ICU, PCR positive. Horizontal black lines indicate LFRET/ELISA cutoffs. SP = spike glycoprotein. NP = nucleoprotein. LFRET = protein L–based time-resolved Förster resonance energy transfer immunoassay. ELISA = enzyme immunoassay. *ρ* = Spearman’s rank correlation coefficient.

**Table 1 viruses-13-00143-t001:** Sensitivity and specificity of ELISA and LFRET in detection of SARS-CoV-2 -PCR-positive individuals for all patients and hospitalized patients at different time points from symptom onset. Absolute numbers of samples are stated after the percentages. Altogether, 19 samples from nine PCR-positive individuals for whom the onset of symptoms is unknown are excluded. NP = nucleoprotein. SP = spike glycoprotein. LFRET = protein L–based time-resolved Förster resonance energy transfer immunoassay. ELISA = enzyme-linked immunosorbent assay.

Patient Group	All Samples	>8 Days after Onset	>13 Days after Onset	All Samples	>8 Days after Onset	>13 Days after Onset
	SP LFRET sensitivity (specificity 100% (52/52))			SP IgG ELISA sensitivity (specificity 100% (52/52))		
All patients	86% (50/58)	92% (49/54)	95% (38/40)	90% (52/58)	94% (51/54)	100% (40/40)
Hospitalized	83% (24/29)	92% (23/25)	100% (13/13)	83% (23/29)	92% (23/25)	100% (13/13)
	NP LFRET sensitivity (specificity 100% (52/52))			NP IgG ELISA sensitivity (specificity 94% (49/52))		
All patients	76% (44/58)	81% (44/54)	80% (32/40)	93% (54/58)	98% (53/54)	100% (40/40)
Hospitalized	79% (23/29)	92% (23/25)	100% (13/13)	86% (25/29)	96% (24/25)	100% (13/13)
	SP/NP LFRET sensitivity (specificity 100% (52/52))			SP/NP IgG ELISA sensitivity (specificity 94% (49/52))		
All patients	90% (52/58)	94% (51/54)	95% (38/40)	93% (54/58)	98% (53/54)	100% (40/40)
Hospitalized	86% (25/29)	96% (24/25)	100% (13/13)	86% (25/29)	96% (24/25)	100% (13/13)

## Data Availability

The data presented in this study are included in the manuscript and the supplementary data.

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
