# Peer review of "A 10-Minute “Mix and Read” Antibody Assay for SARS-CoV-2"

_viruses, 2021, doi:10.3390/v13020143_

Round 1
Reviewer 1 Report
The FRET methodology presented is very helpful but presently still limited to especially equipped laboratories. The method has been compared with an in-house antibody test. However, I suggest that a series of samples should also be compared with at least one of the major commercially available tests.
Generally the work is well done and well documented. It promotes the diagnostic development and it is highly relevant, well suited for "viruses". Fig 1: The size is unusually large; better reduce the size of this figure.
I suggest the publication of this report with minor modifications (a small additional test series for comparison with an established, commercial antibody test
Reviewer 2 Report
The authors describe the development and evaluation of a LFRET (protein L-based time-resolved Förster resonance energy transfer immunoassay) in comparison to ELISA and micro neutralization assay for the detection of specific antibodies directed against SARS-2 CoVit pike (SP) and nucleoprotein 70(NP). The evaluation is well performed. However as already stated by the authors the assay might have a lower analytical sensitivity compared to the ELISA. Since the assay was also not examine for potentially cross-reacting antibodies against the widely circulating corona-viruses OC43, HKU1, NL63 and 229E the evaluation describe more a pilot study than an extensive assay evaluation necessary for a commercial assay. Regardless the preliminary results the novel and faster performance of the assay justify the publication of the results.
Some minor suggestion for changes:
Page 4, line 162, line 168
>3 × 106 cells/ml >3 × 106 cells/ml
Page 6,
line 240. 87% and 100% & 78% and 100%
87% and 100% & 87% and 100%
This is contrast to the numbers in Table 1.
Table 1.: It would be good if the real number of samples used for the calculation of sensitivity and specificity are listed instead of the additional information provided.
“Altogether twenty samples from ten PCR-positive individuals for whom the onset of symptoms is unknown are excluded.”
Table S1: MNT = microneutralization titer.
Abbreviation not used. Please delete
